# Compressed Video Prompt Tuning

**Bing Li**[1,2]    **Jiaxin Chen**[2]    **Xiuguo Bao**[3]    **Di Huang**[1,2*]

[1]SKLSDE, Beihang University, Beijing, China
[2]IRIP Lab, SCSE, Beihang University, Beijing, China
[3]CNCERT/CC, Beijing, China
{libingsy, jiaxinchen, dhuang}@buaa.edu.cn,  baoxiuguo@139.com

## Abstract

Compressed videos offer a compelling alternative to raw videos, showing the possibility to significantly reduce the on-line computational and storage cost. However, current approaches to compressed video processing generally follow the resource-consuming pre-training and fine-tuning paradigm, which does not fully take advantage of such properties, making them not favorable enough for widespread applications. Inspired by recent successes of prompt tuning techniques in computer vision, this paper presents the first attempt to build a prompt based representation learning framework, which enables effective and efficient adaptation of pre-trained raw video models to compressed video understanding tasks. To this end, we propose a novel prompt tuning approach, namely Compressed Video Prompt Tuning (CVPT), emphatically dealing with the challenging issue caused by the inconsistency between pre-training and downstream data modalities. Specifically, CVPT replaces the learnable prompts with compressed modalities (*e.g.* Motion Vectors and Residuals) by re-parameterizing them into conditional prompts followed by layer-wise refinement. The conditional prompts exhibit improved adaptability and generalizability to instances compared to conventional individual learnable ones, and the Residual prompts enhance the noisy motion cues in the Motion Vector prompts for further fusion with the visual cues from I-frames. Additionally, we design Selective Cross-modal Complementary Prompt (SCCP) blocks. After inserting them into the backbone, SCCP blocks leverage semantic relations across diverse levels and modalities to improve cross-modal interactions between prompts and input flows. Extensive evaluations on HMDB-51, UCF-101 and Something-Something v2 demonstrate that CVPT remarkably outperforms the state-of-the-art counterparts, delivering a much better balance between accuracy and efficiency.

## 1  Introduction

In recent years, there has been a surge of interest in compressed video understanding [51, 36, 8, 46, 11, 19] due to its great potential in saving the computational and storage cost for on-line processing, which directly performs inference on compressed data and thus bypasses the resource-consuming decoding phase.

Raw videos consist of dense RGB frames conveying coherent and consistent content, while compressed videos are composed of sparsely decoded RGB frames with entire spatial information, *i.e.* Intra-frames (I-frames), and incompletely decoded RGB frames with free but coarse motion clues, *i.e.* Predicted frames (P-frames) or Bidirectionally predicted frames (B-frames). In this case, existing approaches to compressed video understanding generally follow the pre-training and fine-tuning paradigm, which initially pre-trains models on compressed videos from a large database (*e.g.* Kinetics-400 [3]) and fully fine-tunes them on downstream ones. According to different network structures,

---

*Corresponding author.

37th Conference on Neural Information Processing Systems (NeurIPS 2023).

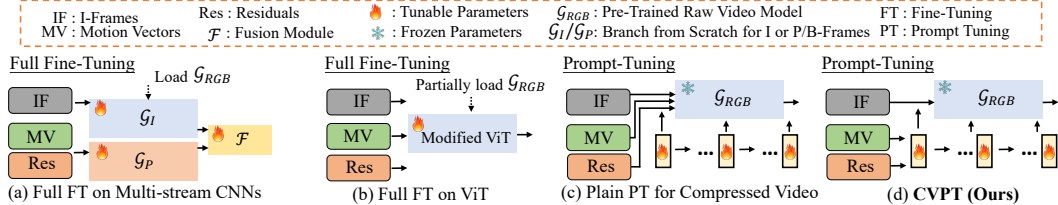

Figure 1: Different types of approaches to representation learning in compressed videos. (a) and (b) are the two main variants of the existing pre-training and fine-tuning paradigm, where the former is based on multi-stream CNNs and the latter is based on ViT, both of which require pre-training models on compressed videos. (c) is a plain prompt tuning scheme which can be considered as a straightforward attempt to leverage frozen off-the-shelf models pre-trained on raw videos, but it does not well adapt to the given task with degraded performance. (d) CVPT advances (c) by designing conditional prompts for Motion Vectors and Residuals, delivering a novel paradigm which successfully learns powerful representations for compressed videos in a more effective and efficient manner.

they can be broadly classified into two categories. (i) [20, 1, 21, 19, 11] apply multi-branch CNNs for individual modalities, which are then fused to deliver final results, as depicted in Figure 1 (a). (ii) [5] employs a modified Vision Transformer (ViT) for all modalities, motivated by the advantage of ViT in reasoning tasks and the ability to handle multi-modalities in text-image [15, 37, 50] and multi-task domains [31, 18], as shown in Figure 1 (b). Despite their effectiveness, such a paradigm suffers two major drawbacks. Firstly, it requires pre-training networks on compressed videos, resulting in a high computational complexity in GPU-day in off-line mode. Several attempts are made to speed up pre-training by loading or partially loading off-the-shelf models built based on raw videos; however, it is still not favorable enough for wide popularization. Secondly, it relies on fully fine-tuning backbones, which incurs a large parameter storage burden, making it challenging for deployment in real-world applications. Furthermore, full fine-tuning tends to disrupt the knowledge learned by the pre-trained model, degrading the generalization performance [54]. The limitations aforementioned highlight the need for a more effective and efficient alternative to address these issues.

Inspired by the advancements of prompting techniques in Natural Language Processing (NLP), researchers from the field of Computer Vision (CV) have been discussing this representation learning paradigm. By freezing the pre-trained model and introducing a small number of learnable parameters on input images or videos, they create decent visual prompts [4, 23]. This paradigm enables efficient information mining from pre-trained models and eliminates the necessity for fully fine-tuning networks. On the other side, raw videos contain richer spatial and temporal clues than compressed ones, and if sufficient and important knowledge can be mined from the models pre-trained on raw videos, it should strengthen representation learning of compressed videos. Both the facts trigger us to investigate a prompt tuning paradigm to compressed video understanding with pre-trained raw video models as a more effective and efficient alternative to the pre-training and fine-tuning one. Although current prompt tuning approaches [32, 16, 24, 4, 9] have achieved successes in various tasks, they generally work in the case that pre-training and downstream data share the same modality and it remains problematic when they have different modalities, *i.e.* the pre-training data are in the raw video (dense RGB frames) modality and the downstream data are in the compressed video modality (I-frames, Motion Vectors and Residuals), as shown in Figure 1 (c).

In this paper, we present a novel paradigm to video understanding in the compressed domain, namely Compressed Video Prompt-Tuning (CVPT). As in Figure 1 (d), our approach freezes the whole pre-trained raw video-based model and learns a few modal-specific visual prompts. It allows us to leverage the common prior knowledge in the feature and semantic spaces shared by the two types of videos, to facilitate representation learning of compressed videos. Specifically, VPT employs a conditional prompt framework with refinement, in which Motion Vectors and Residuals are re-parameterized into conditional prompts. These prompts are more adaptable to individual instances and alleviate the impact of freezing these modalities which can lead to performance degradation. Additionally, CVPT incorporates simple and lightweight modality-complementary prompt blocks, namely Selective Cross-modal Complementary Prompt (SCCP), into the frozen pre-trained model, capturing cross-modal interactions for performance improvement. Concretely, SCCP firstly embeds the I-frame

tokens as well as the conditional Motion Vector and Residual prompts into a low-dimensional feature space, in pursuit of computational efficiency. Based on attention maps induced by the Residual prompts, the Motion Vector prompts are subsequently refined via suppressing the noisy motions, which are further integrated into the I-frame tokens as complementary motion cues. As a consequence, the cross-modal interaction is significantly strengthened, remarkably facilitating model tuning for efficient representation learning of compressed videos.

The main contributions of this paper are summarized in four-fold:

- We propose CVPT, a novel visual prompt tuning framework, which enables pre-trained raw video models to adapt to compressed video understanding tasks. To the best of our knowledge, this is the first work to address this issue.

- We introduce modal-related conditional prompts to replace conventional learnable ones, proving more adaptable to individual instances and alleviates the inconsistency between pre-training and downstream data modalities.

- We design selective cross-modal complementary prompt blocks, and these blocks substantially strengthen cross-modal interactions and selectively complement dynamic cues in Motion Vectors to input flows.

- We conduct extensive experiments on three widely used benchmarks and achieve newly state-of-the-art results. Notably, our approach maintains the parameter-efficiency with less than 1% parameters being trainable.

## 2 Related Work

### 2.1 Compressed Video Representation Learning

Early studies commonly adopt multi-stream CNNs and focus on dedicatedly designing the model structures for P/B-frames and fusion modules. The pioneering work by [49] introduces the use of CNNs to process compressed videos. They replace optical flows in the two-stream network [14] with Motion Vectors, resulting in a higher efficiency. CoViAR [45] extends a third stream to leverage all the modalities, including I-frames, Motion Vectors, and Residuals, completely bypassing video decoding. To improve the quality of dynamics, [2] applies image denoising techniques, while [40] employs GANs to generate Motion Vectors that approximate optical flows. [22] and [1] adopt a similar approach by feeding compressed modalities into a CNN model to mimic a raw video-based teacher. [22] presents Temporal Trilinear Pooling for fusing multiple modalities in lightweight models suitable for portable devices. Inspired by SlowFast [12, 30] propose the Slow-I-Fast-P model, where pseudo optical flows are estimated by a specific loss function. MM-ViT [5] is the first work using ViT on compressed videos, factorizing the self-attention across the space, time and modality dimensions.

The advent of self-supervised learning, particularly contrastive learning and Masked AutoEncoder (MAE), has led to the exploration of self-supervised pretext tasks for compressed videos. In this context, IMRNet [48] stands out as the first work to delve into self-supervised learning for compressed videos. It employs a three-stream network to individually encode I-frames, Motion Vectors, and Residuals, and introduces two pretext tasks that make use of motion statistics and internal GOP (Group of Pictures) structures in the compressed domain. Another notable approach is MVCGC [21], which delivers a cross-guidance contrastive learning framework with multi-views. MVCGC utilizes Motion Vectors as supervision signals for RGB frames and vice versa.

To sum up, the existing approaches in compressed video understanding involve the design of specialized modules or pretext tasks, which makes the pre-training and fine-tuning paradigm resource-consuming, not favorable enough to various downstream applications. In contrast, our work presents a prompt tuning based alternative, where necessary knowledge are mined from the off-the-shelf model pre-trained on raw videos through only a few modal-specific visual prompts, and is more effective and efficient.

### 2.2 Visual Prompt Tuning

Prompt tuning has emerged as a promising way to efficient representation learning in the field of NLP. By leveraging textual prompts, researchers have successfully reformulated downstream tasks

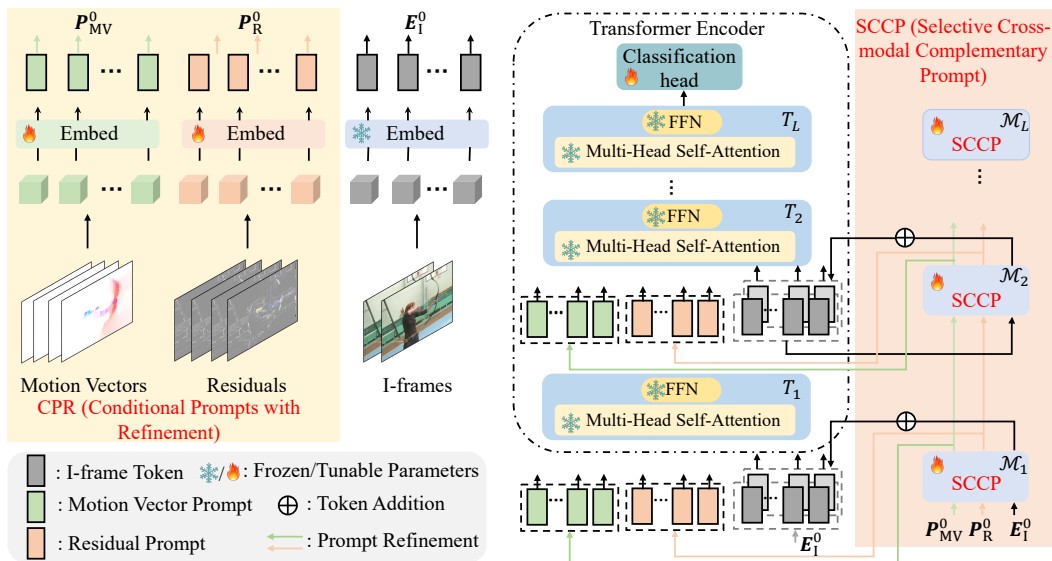

Figure 2: Framework of the proposed CVPT. It firstly embed the input compressed videos, consisting of I-frames, Motion Vectors and Residuals, into the I-frame tokens $E_{\mathrm{I}}^0$, the conditional Motion Vector prompts $P_{\mathrm{MV}}^0$ and the Residual prompts $P_{\mathrm{R}}^0$, respectively. A frozen pre-trained Transformer with $L$ layers is adopt for feature extraction. The Selective Cross-modal Complementary Prompt (SCCP) blocks are employed to perform conditional prompt learning as well as facilitate the interactions between the prompts and the I-frame tokens for efficient fine-tuning.

to resemble pre-training tasks that have already been solved [28, 33]. This approach offers several advantages over full fine-tuning, such as the high parameter efficiency and the reduced storage cost. It has also shown promising results in various CV applications. For instance, VPT [23] introduces learnable parameters to Transformer encoders and achieves superior performance compared to full fine-tuning on multiple recognition tasks. AdaptFormer [7] enhances ViT models with lightweight modules and surpasses the performance of fully fine-tuned models in action recognition benchmarks. Convpass [25] proposes convolutional bypasses for prompt learning in pre-trained ViT models. Some prompting attempts have also been made to deal with multi-modal tasks, such as visual and language. CoOp [53] fine-tunes CLIP using a continuous set of prompt vectors, while Co-CoOp [52] improves its generalization by conditioning prompts on image instances. [26] strengthens CLIP for video understanding tasks by adding lightweight Transformers on visual features and text prompts. MaPLe [27] optimizes the prompting mechanism in both the vision and language branches to improve the adaptability of CLIP.

Different from current prompt-learning approaches in NLP and CV that primarily apply in the case that pre-training and downstream data share the same modality, our work further addresses the challenge arising from the inconsistency between pre-training and downstream data modalities, *i.e.* the pre-training data are dense RGB frames while the downstream data are sparse I-frames and coarse Motion Vectors and Residuals.

## 3 The Proposed Approach

In this section, we describe the details of the proposed Compressed Video Prompt Tuning (CVPT) approach. Briefly, CVPT employs a conditional prompt framework with refinement as shown in Figure 2, by proposing a Selective Cross-modal Complementary Prompt (SCCP), which are depicted in the rest part.

### 3.1 Conditional Prompt with Refinement

Prompt tuning [28, 33] is commonly utilized to efficiently fine-tune a model pre-trained on the upstream tasks to align with the downstream tasks. Existing prompt tuning approaches often assume

that the pre-trained task and the downstream task share the same data modality. However, when fine-tuning for compressed video-based applications, the data modality differs as the downstream tasks adopt the compressed videos, and the pre-trained tasks utilize the raw videos. A straightforward fine-tuning without considering the modality gap often severely deteriorates the performance, since the Motion Vectors and Residuals are unique to compressed videos and have distinct data distributions from the raw videos. To address this issue, existing works employ an extra feature extraction network that may negatively abate the inter-modal interactions; or modify the Transformer structure to process the multi-modal compressed video data, which however still requires pre-training and full fine-tuning.

In this paper, we propose an efficient prompt tuning approach for compassed video-based tasks. Motivated by [54], we re-parameterize the Motion Vectors and Residuals into prompts and enhance the cross-modal interactions with I-frames for prompt learning. Different from [54], we embed the Motion Vectors and Residuals into distinct types of prompts, and refine the Motion Vector prompts by leveraging the Residual counterparts, which are further integrated into the I-frame Motion Vectors as strengthened complements of motion cues.

Concretely, the input I-frames $\boldsymbol{X}_{\mathrm{I}} \in \mathbb{R}^{3 \times T_I \times H \times W}$ are firstly encoded into the I-frame tokens $\boldsymbol{E}_{\mathrm{I}}^0 \in \mathbb{R}^{N \times d}$ by a frozen patch embedding layer, where $H$, $W$, $d$ represent the frame height, frame width and the encoding dimension, respectively. The Motion Vectors $\boldsymbol{X}_{\mathrm{MV}} \in \mathbb{R}^{2 \times T_{\mathrm{P}} \times H \times W}$ and the Residuals $\boldsymbol{X}_{\mathrm{R}} \in \mathbb{R}^{3 \times T_P \times H \times W}$ are encoded by learnable patch embedding layers, generating the conditional Motion Vector prompts $\boldsymbol{P}_{\mathrm{MV}}^0 \in \mathbb{R}^{N \times d}$ and the Residual prompts $\boldsymbol{P}_{\mathrm{R}}^0 \in \mathbb{R}^{N \times d}$, respectively. $T_{\mathrm{I}}/T_{\mathrm{P}}$ denote the number of I/P-frames.

The I-frame tokens $\boldsymbol{E}_{\mathrm{I}}^0$ together with the conditional prompts $\boldsymbol{X}_{\mathrm{MV}}$ and $\boldsymbol{X}_{\mathrm{R}}$ are comprehensively integrated by the proposed Selective Cross-modal Complementary Prompt (SCCP) at the first layer denoted by $\mathcal{M}_1(\cdot)$, where the output are fed into the Transformer block as complementary motion cues to the I-frame tokens, as well as inputs to the SCCP block from the subsequent layer. The SCCP blocks are successively stacked by aligning with the original ViT layers. The overall forward propagation process for prompt tuning can be briefly formulated as follows:

$$[\boldsymbol{P}_{\mathrm{MV}}^l, \boldsymbol{P}_{\mathrm{R}}^l, \boldsymbol{E}_{\mathrm{I}}^{l-1}] = T_l\left(\left[\hat{\boldsymbol{P}}_{\mathrm{MV'}}^{l-1}, \hat{\boldsymbol{P}}_{\mathrm{R'}}^{l-1}, \boldsymbol{E}_{\mathrm{I}}^{l-1} + \hat{\boldsymbol{E}}_{\mathrm{I'}}^{l-1}\right]\right), \qquad l = 1, 2, ...L; \qquad (1)$$

$$\boldsymbol{y} = \mathrm{CLS\_Head}\left(\left[\boldsymbol{P}_{\mathrm{MV}}^L, \boldsymbol{P}_{\mathrm{R}}^L, \boldsymbol{E}_{\mathrm{I}}^L\right]\right). \qquad (2)$$

where $L$ is the number of Transformer layers, $T_l(\cdot)$ and $\mathrm{CLS\_Head}(\cdot)$ refer to the Transformer block at the $l$-th layer and the classification head, respectively. Here, $[\hat{\boldsymbol{P}}_{\mathrm{MV'}}^{l-1}, \hat{\boldsymbol{P}}_{\mathrm{R'}}^{l-1}, \hat{\boldsymbol{E}}_{\mathrm{I'}}^{l-1}]$ represents the output of the SCCP block at the $l$-th layer, which will be elaborated in Section 3.2.

### 3.2 Selective Cross-modal Complementary Prompt

Given the I-frame tokens $\boldsymbol{E}_{\mathrm{I}}^0$, the conditional Motion Vector prompts $\boldsymbol{P}_{\mathrm{MV}}^0$, the Residual prompts $\boldsymbol{P}_{\mathrm{R}}^0$, and a frozen pre-trained encoder $\mathcal{G}_{RGB}$ containing $L$ Transformer layers, SCCP aims to complement the motion cues from the conditional prompts to the I-frame branch. Generally, SCCP aggregates the tokens and prompts from multiple modalities of the previous layer to generate the new ones at the current layer, which is formulated as below:

$$\left[\boldsymbol{P}_{\mathrm{MV'}}^{l-1}, \boldsymbol{P}_{\mathrm{R'}}^{l-1}, \boldsymbol{E}_{\mathrm{I'}}^{l-1}\right] = \mathcal{M}_l\left(\left[\boldsymbol{P}_{\mathrm{MV}}^{l-1}, \boldsymbol{P}_{\mathrm{R}}^{l-1}, \boldsymbol{E}_{\mathrm{I}}^{l-1}\right]\right), \qquad l = 1, 2, ...L. \qquad (3)$$

It is worth noting that the Motion Vectors, which are block-wise motion cues estimated by matching, tend to be coarse and noisy compared to the optical flow counterparts displayed in Figure 3(a). As depicted in [39], motion information near object boundaries plays a crucial role in representation learning. Inspired by this insight, we generate an attention map to select refined motion cues from the Motion Vectors by leveraging the Residuals, based on the observation that the Residuals are generally well-aligned with the boundaries of moving objects and strongly correlated with Motion Vectors as observed in [40]. The refined motion cues are successively adopted as a informative complements to the RGB modality, thus boosting learning discriminative representations.

The detailed design of SCCP is illustrated in Figure 3(b). Specifically, at the $l$-th layer, the input of SCCP consists of the I-frame tokens $\boldsymbol{E}_{\mathrm{I}}^{l-1}$, the conditional prompts $\boldsymbol{P}_{\mathrm{MV}}^{l-1}$ and $\boldsymbol{P}_{\mathrm{R}}^{l-1}$. In order to obtain compact representations and reduce the overall size of parameters, we firstly embed the $C$-dimensional input into a $(C/\alpha)$-dimensional feature space as follows:

$$\hat{\boldsymbol{E}}_{\mathrm{I}}^{l-1} = g_1(\boldsymbol{E}_{\mathrm{I}}^{l-1}), \quad \hat{\boldsymbol{P}}_{\mathrm{MV}}^{l-1} = g_2(\boldsymbol{P}_{\mathrm{MV}}^{l-1}), \quad \hat{\boldsymbol{P}}_{\mathrm{R}}^{l-1} = g_3(\boldsymbol{P}_{\mathrm{R}}^{l-1}), \qquad (4)$$

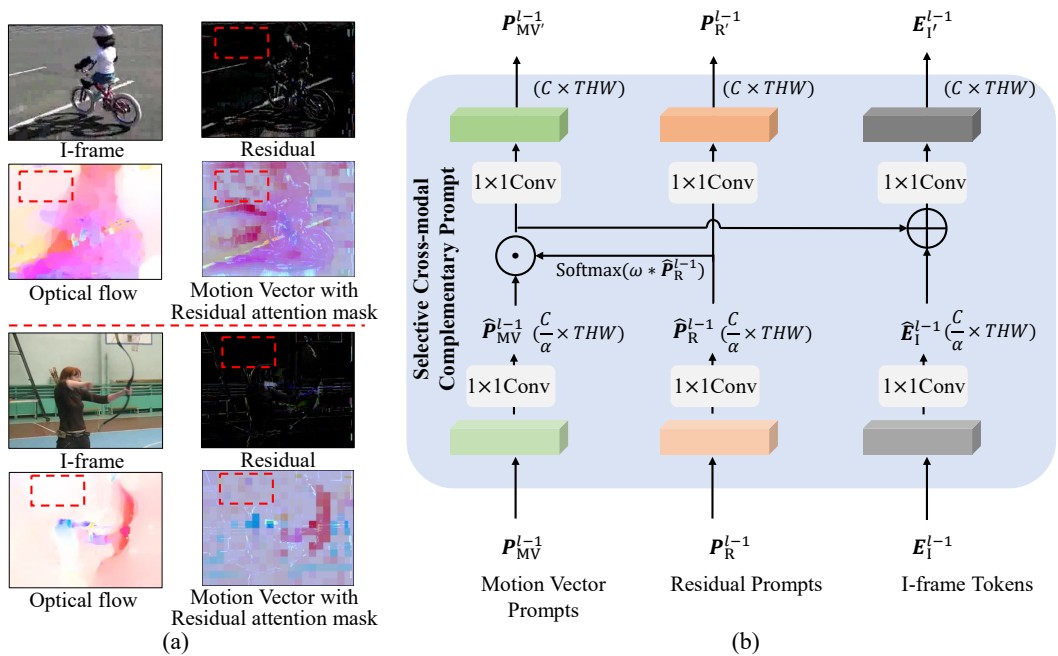

Figure 3: (a) Illustration on the optical flows representing accurate motion cues, the coarse Motion Vectors and the Residuals. The red rectangles highlight the areas that Motion Vectors contain noises, which however can be refined by leveraging the Residuals. (b) Detailed architecture of the proposed Selective Cross-modal Complementary Prompt.

where $\alpha$ is the reduction factor, the embedding functions $g_1(\cdot)$, $g_2(\cdot)$, and $g_3(\cdot)$ are simple $1 \times 1$ convolutions. In the ViT-B backbone, the input feature dimension $C$ is 768, and the reduction factor $\alpha$ is set to 96 in all prompt blocks. As the scale of backbone grows, $\alpha$ usually increases proportionally. In the case of Swin Transformer, as the feature dimension $C$ becomes larger in deeper layers, we gradually increase it to maintain the same embedding feature size across different layers.

After feature embedding, we generate a spatial-temporal attention map $\mathcal{A}$ by performing $\mathrm{Softmax}(\cdot)$ on the embedded Residual prompts $\hat{\boldsymbol{P}}_{\mathrm{R}}^{l-1}$. To strengthen the smoothness of the attention map, we introduce a learnable parameter. The attention map is subsequently applied to select representative motion cues from the embedded Motion Vector prompts $\hat{\boldsymbol{P}}_{\mathrm{MV}}^{l-1}$ as the following:

$$\hat{\boldsymbol{P}}_{\mathrm{MV}}^{l-1} = \mathcal{A} \odot \hat{\boldsymbol{P}}_{\mathrm{MV}}^{l-1} = \mathrm{Softmax}(\omega \times \hat{\boldsymbol{P}}_{\mathrm{R}}^{l-1}) \odot \hat{\boldsymbol{P}}_{\mathrm{MV}}^{l-1}. \tag{5}$$

The refined prompts $\hat{\boldsymbol{P}}_{\mathrm{MV}}^{l-1}$ is further integrated into the intermediate embedding $\hat{\boldsymbol{E}}_{\mathrm{I}}^{l-1}$ by addition, in order to complement the motion cues as below

$$\hat{\boldsymbol{E}}_{\mathrm{I}}^{l-1} := \hat{\boldsymbol{E}}_{\mathrm{I}}^{l-1} + \hat{\boldsymbol{P}}_{\mathrm{MV}}^{l-1}. \tag{6}$$

After performing the above cross-modal interactions, the embedded tokens $\hat{\boldsymbol{E}}_{\mathrm{I}}^{l-1}$, the embedded prompts $\hat{\boldsymbol{P}}_{\mathrm{MV}}^{l-1}$ and $\hat{\boldsymbol{P}}_{\mathrm{R}}^{l-1}$ are finally project back to the high-dimensional ones by using the following formulation:

$$\hat{\boldsymbol{E}}_{\mathrm{I}'}^{l-1} = g_4(\hat{\boldsymbol{E}}_{\mathrm{I}}^{l-1}), \quad \hat{\boldsymbol{P}}_{\mathrm{MV}'}^{l-1} = g_5(\hat{\boldsymbol{P}}_{\mathrm{MV}}^{l-1}), \quad \hat{\boldsymbol{P}}_{\mathrm{R}'}^{l-1} = g_6(\hat{\boldsymbol{P}}_{\mathrm{R}}^{l-1}), \tag{7}$$

where $g_4(\cdot)$, $g_5(\cdot)$, and $g_6(\cdot)$ are $1 \times 1$ convolutions.

In order to comprehensively enhance the representation learning, the SCCP block is applied to each Transformer block, and is successively stacked, *i.e.* the conditional prompts $\boldsymbol{P}_{\mathrm{MV}}^{l-1}$ and $\boldsymbol{P}_{\mathrm{R}}^{l-1}$ are updated in each SCCP block and is fed into the next SCCP block as input.

# 4 Experimental Results and Analysis

## 4.1 Datasets and Evaluation Metric

**HMDB-51** and **UCF-101** are two relatively small datasets, which contain 6766 videos from 51 action categories and 13,320 videos from 101 categories, respectively. **Something-Something v2** (SSv2) is a large-scale motion-centric video dataset, including 168,913 videos for training and 24,777 videos for validation from 174 categories. As a typical video understanding task, we choose action recognition and report top-1 accuracy as the evaluation metric in all experiments.

## 4.2 Implementation Details

By following [45, 29], we convert the raw videos into the MP4 compressed format with a GOP size of 12, where each GOP consists of 1 I-frame and 11 P-frames. We collect I-frames, Motion Vectors and Residuals from 4 consecutive GOPs, totally containing 4 I-frames and 12 P-frames. Within each GOP, we uniformly sample 3 P-frames. In regards of the backbone network, we utilize two representative video Transformer architectures based on ViT [10] and Swin Transformer [34] pre-trained on Kinetics-400 with raw videos, respectively. We adopt the original model configurations and train the prompt parameters using the AdamW optimizer [35] on 12 NVIDIA V100 GPUs. We apply random scaling, corner cropping, and horizontal flipping as data augmentation. The base learning rate, weight decay and batch size are set to $1 \times 10^{-3}$, $1 \times 10^{-4}$ and 240, respectively. Additionally, we adopt a warm-up strategy within the first 5 training epochs.

## 4.3 Comparison with the State-of-the-art Approaches

**Compressed video-based methods.** We firstly compare our method with the following full fine-tuning compressed video-based models: 1) self-supervised pre-training based ones including CoViAR [45], IMRNet [48], MVCGC [21] and the Full Fine-tuning baseline; and 2) supervised pre-training based ones including Refined-MV [2], IPTSN [20], SIFP-Net [30], MEACI-Net [29] and MM-ViT [5]. Table 1 and Table 2 summarize the comparison results on the HMDB-51, UCF-101 and SSv2 datasets. Generally, the proposed CVPT method achieves comparable or even better performance than the full fine-tuning ones, by tuning less than 1% of the overall parameters. Notably, when adopting the same ViT-B backbone, CVPT improves the top-1 accuracy of MM-ViT/Full Fine-tuning by 2.1%/0.9% on UCF-101, and 2.9%/0.8% on SSv2, via tuning only 0.6% of the parameters.

Besides the full fine-tuning methods, we also compare with representative efficient fine-tuning models, including the Linear Probe baseline, VPT [23] and AdaptFormer [7]. Since VPT and AdaptFormer only report results on raw videos, we re-implement and evaluate them on compressed videos, denoted by VPT* and Adaptformer*, respectively. As displayed, Linear Probe exhibits the lowest performance as it only optimizes the classification head. VPT* slightly outperforms Adaptformer*, but both of their performance are clearly inferior to the fine-tuning counterparts. This suggests that these methods struggle to effectively align the upstream data for pre-training the downstream data for fine-tuning. In contrast, our approach leverages inserted prompt blocks to facilitate the interaction of multimodal information, boosting the accuracy of VPT* and Adaptformer* by 10.8% and 12.0% on HMDB-51, 5.4% and 7.1% on UCF-101, and 4.8% and 5.6% on SSv2, respectively, when pre-training with the self-supervised learning.

In addition to the accuracy, our approach also has advantages in its training efficiency and facilitation. Concretely, unlike IPTSN that adopts a complex knowledge distillation model based on a raw video-based teacher during fine-tuning, or MVCGC that mines cross-modality information from pre-trained RGB models through self-supervised learning, our method explores the efficient tuning and reaches a promising accuracy. Moreover, as shown in Table 1 and Table 2, our approach consistently promotes the performance by different pre-training strategies and network architectures (*e.g.* ViT-L), highlighting its scalability.

**Raw video-based methods.** As summarized in Table 1 and Table 2, raw video-based fine-tuning methods generally reach higher performance than compressed video-based counterparts, since the raw videos convey more coherent and consistent information. Nevertheless, CVPT significantly reduces this gap, and even surpasses both VPT and Adaptformer that are fine-tuned on raw videos. The reason lies in that CVPT explores the Motion Vectors and Residuals as prompts to complement the missing motion cues caused by sparse sampling of RGB frames in a more comprehensive way.

Table 1: Comparison with the state-of-the-art approaches w.r.t. the top-1 accuracy (%) on HMDB-51 and UCF-101 pre-trained on Kinetics-400. '∗' indicate the results by our re-implementation using the same configuration. 'PT' stands for pre-training. 'SL' and 'SSL' refer to supervised learning and self-supervised learning, respectively.

| | Method | Modality | Backbone | Input [M] | GFLOPs | PT | Tunable Params. [M] | HMDB-51 | UCF-101 |
|---|---|---|---|---|---|---|---|---|---|
| **Raw** | CoCLR [17] | RGB+OF | S3D | 26.2 | 182.0 | SSL | 17.5 (100%) | 62.9 | 90.6 |
| | RSPNet [6] | RGB | S3D-G | 6.0 | 91.0 | SSL | 9.6 (100%) | 64.7 | 93.7 |
| | CVRL [38] | RGB | R3D-50 | 48.2 | 402.0 | SSL | 46.9 (100%) | 65.4 | 92.1 |
| | $\rho$BYOL [13] | RGB | R3D-50 | 12.1 | 402.0 | SSL | 46.9 (100%) | 73.6 | 95.5 |
| | STS [43] | RGB | S3D-G | 96.3 | 689.0 | SSL | 9.6 (100%) | 62.0 | 89.0 |
| | M$^3$Video [41] | RGB | ViT-B | 24.1 | 1080.0 | SSL | 86.4 (100%) | 75.4 | 96.1 |
| | MVD-B [44] | RGB | ViT-B | 24.1 | 1080.0 | SSL | 86.4 (100%) | 76.4 | 97.0 |
| | MotionMAE [47] | RGB | ViT-B | 24.1 | 1080.0 | SSL | 86.4 (100%) | - | 96.3 |
| | VideoMAE [42] | RGB | ViT-B | 24.1 | 1080.0 | SSL | 86.4 (100%) | 73.3 | 96.1 |
| | VPT [23] | RGB | ViT-B | 12.1 | 1089.6 | SSL | 0.1 (0.09%) | 52.7 | - |
| | AdaptFormer [7] | RGB | ViT-B | 12.1 | 1093.8 | SSL | 1.3 (1.46%) | 55.7 | - |
| **Comp.** | CoViAR [45] | I-Frame+MV+Res | ResNet152 | 3.8 | 1222.0 | SSL | 142.5 (100%) | 37.1 | 63.7 |
| | IMRNet [48] | I-Frame+MV+Res | R3D-18 | 3.8 | - | SSL | 100.8 (100%) | 45.0 | 76.8 |
| | MVCGC [21] | RGB+MV | S3D | 15.7 | - | SSL | 17.5 (100%) | 63.4 | 90.8 |
| | Refined-MV [2] | I-Frame+MV+Res | ResNet152 | - | - | SL | 142.5 (100%) | 59.7 | 89.9 |
| | IPTSN [20] | I-Frame+MV+Res | ResNet152 | 6.8 | - | SL | 130.8 (100%) | 69.1 | 93.4 |
| | SIFP-Net [30] | I-Frame+MV+Res | I3D | 8.1 | - | SL | 92.8 (100%) | 72.3 | 94.0 |
| | MEACI-Net [29] | I-Frame+MV+Res | I3D | 0.7 | 269.4 | SL | 50.7 (100%) | 74.4 | 96.4 |
| | MM-ViT [5] | I-Frame+MV+Res | ViT-B | 8.1 | - | SL | 86.4 (100%) | - | 93.4 |
| | Full Fine-tuning | I-Frame+MV+Res | ViT-B | 6.1 | 772.2 | SSL | 86.4 (100%) | 60.3 | 88.1 |
| | Linear Probe | I-Frame+MV+Res | ViT-B | 6.1 | 772.2 | SSL | 0.1 (0.08%) | 47.8 | 76.8 |
| | VPT∗ [23] | I-Frame+MV+Res | ViT-B | 6.1 | 778.2 | SSL | 0.1 (0.09%) | 52.1 | 83.6 |
| | AdaptFormer∗ [7] | I-Frame+MV+Res | ViT-B | 6.1 | 780.6 | SSL | 1.3 (1.46%) | 50.9 | 81.9 |
| | **CVPT (Ours)** | I-Frame+MV+Res | ViT-B | 6.1 | 772.2 | SSL | 0.5 (0.6%) | 62.9 | 89.0 |
| | **CVPT (Ours)** | I-Frame+MV+Res | ViT-B | 6.1 | 772.2 | SL | 0.5 (0.6%) | 69.7 | 95.5 |
| | **CVPT (Ours)** | I-Frame+MV+Res | ViT-L | 6.1 | 2569.6 | SL | 0.1 (0.03%) | 81.5 | 98.0 |

Table 2: Comparison with the state-of-the-art approaches w.r.t. the top-1 accuracy (%) on SSv2 pre-trained on Kinetics-400. '∗' indicate the results by our re-implementation using the same configuration. 'PT' stands for pre-training. 'SL' and 'SSL' refer to supervised learning and self-supervised learning, respectively.

| | Method | Modality | Backbone | Input [M] | GFLOPs | PT | Tunable Params. [M] | SSv2 |
|---|---|---|---|---|---|---|---|---|
| **Raw** | RSPNet [6] | RGB | S3D-G | 6.0 | 91.0 | SSL | 9.6 (100%) | 55.0 |
| | $\rho$BYOL [13] | RGB | R3D-50 | 12.1 | 402.0 | SSL | 46.9 (100%) | 55.8 |
| | MVD-B [44] | RGB | ViT-B | 24.1 | 1080.0 | SSL | 86.4 (100%) | 72.5 |
| | M$^3$Video [41] | RGB | ViT-B | 24.1 | 1080.0 | SSL | 86.4 (100%) | 70.5 |
| | MotionMAE [47] | RGB | ViT-B | 24.1 | 1080.0 | SSL | 86.4 (100%) | 71.8 |
| | VideoMAE [42] | RGB | ViT-B | 24.1 | 1080.0 | SSL | 86.4 (100%) | 70.3 |
| | VPT [23] | RGB | ViT-B | 12.1 | 1089.6 | SSL | 0.1 (0.09%) | 43.7 |
| | AdaptFormer [7] | RGB | ViT-B | 12.1 | 1093.8 | SSL | 1.3 (1.46%) | 59.0 |
| **Comp.** | MM-ViT [5] | I-Frame+MV+Res | ViT-B | 8.1 | - | SL | 86.4 (100%) | 62.6 |
| | Full Fine-tuning | I-Frame+MV+Res | ViT-B | 6.1 | 772.2 | SSL | 86.4 (100%) | 57.6 |
| | Linear Probe | I-Frame+MV+Res | ViT-B | 6.1 | 772.2 | SSL | 0.1 (0.08%) | 33.8 |
| | VPT∗ [23] | I-Frame+MV+Res | ViT-B | 6.1 | 778.2 | SSL | 0.1 (0.09%) | 53.6 |
| | AdaptFormer∗ [7] | I-Frame+MV+Res | ViT-B | 6.1 | 780.6 | SSL | 1.3 (1.46%) | 52.8 |
| | **CVPT (Ours)** | I-Frame+MV+Res | ViT-B | 6.1 | 772.2 | SSL | 0.5 (0.6%) | 58.4 |
| | **CVPT (Ours)** | I-Frame+MV+Res | ViT-L | 6.1 | 2569.6 | SL | 0.1 (0.03%) | 65.5 |

Additionally, CVPT is capable of directly fine-tuning an off-the-shelf model pre-trained on raw-videos to the domain of compressed videos, thus simultaneously saving extra computational cost in alternatively pre-training on the specific compressed data.

## 4.4 Ablation Study

To evaluate the effectiveness of components in CVPT, we conduct extensive ablation study. Without loss of generality, we utilize ViT-B pre-trained by self-supervised learning as the backbone.

**On the Main Components.** We firstly evaluate the effect of the main components of the proposed method. Specifically, we employ linear probe as the baseline method. As shown in Table 3, the proposed CPR module without refinement, which only updates the conditional prompts in the first layer, clearly improve the baseline. By introducing the SCCP module, which enriches the cross-modal information between I-frames and conditional prompts of Motion Vectors and Residuals, the

Table 3: Ablation results (%) on the main components of CVPT.

| CPR w/o refinement | SCCP | CPR | HMDB-51 | UCF-101 |
|:---:|:---:|:---:|:---:|:---:|
| | | | 47.8 | 76.8 |
| ✓ | | | 55.7 | 83.9 |
| ✓ | ✓ | | 61.4 | 87.6 |
| | ✓ | ✓ | 62.9 | 89.0 |

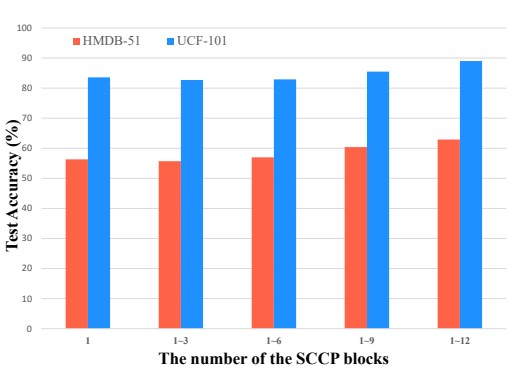

Figure 4: The influence of the number of SCCP blocks on the proposed method.

Figure 5: Illustrations of alternative structures of SCCP. (a) and (b) indicate SCCP without $\boldsymbol{P}_R$ prompt and without attentions, respectively.

performance is remarkably boosted. Finally, applying the refinement module on CPR further obtains gain in performance, leading to the highest accuracy.

**On the Selective Cross-modal Complementary Prompt (SCCP).** To demonstrate the effect of the proposed SCCP module, we compare it to the following two alternative architecture designs as shown in Figure 5: 1) removing the $\boldsymbol{P}_R$ prompt and separately embedding the Motion Vectors and Residuals into distinct two types of prompts as displayed in Figure 5(a); 2) abandoning the Residual-guided attention and directly adding to the branch of Motion Vectors as displayed in Figure 5(b). As summarized in Table 4, both of the two above alternative designs yield worse performance than the proposed SCCP module, thus demonstrating the effectiveness of employing the Residual prompt to facilitate extracting more informative dynamic cues for the downstream task.

Besides, we investigate the impact of incorporating different numbers of SCCP blocks into the backbone. The insertion process begins from the first layer and increments by three layers in each iteration. As illustrated in Figure 4, we observe that the overall performance on both datasets is consistently improved as the number of inserted SCCP blocks increases.

**On Distinct Backbones.** We evaluate the performance of our method by using different backbones, compared to the Full Fine-tuning baseline. As summarized in Table 5, our approach steadily outperforms Full Fine-tuning for various backbones by distinct pre-training strategies.

### 4.5   On the Efficiency of CVPT

We present a comprehensive comparison of inference time between the proposed CVPT method and the representative raw video-based approach VideoMAE. As shown in Table 6, our approach

Table 4: Ablation study (%) on different structures of the proposed SCCP module.

| Method | Total Params. [K] | HMDB-51 | UCF-101 |
|:---:|:---:|:---:|:---:|
| SCCP w/o attention | 39.2 | 59.6 | 86.9 |
| SCCP w/o $\boldsymbol{P}_R$ prompt | 19.2 | 57.6 | 85.5 |
| SCCP | 39.2 | 61.4 | 87.6 |

Table 5: Ablation study (%) on different backbones.

| Backbone | ViT-B (SSL) | | ViT-B (SL) | | Swin-B (SL) | |
|---|---|---|---|---|---|---|
| | HMDB-51 | UCF-101 | HMDB-51 | UCF-101 | HMDB-51 | UCF-101 |
| Full Fine-tuning | 60.3 | 88.1 | 66.5 | 94.3 | 62.9 | 92.0 |
| **Ours** | 62.9 (+2.6) | 89.0 (+0.9) | 69.7 (+3.2) | 95.5 (+1.2) | 64.9 (+2.0) | 94.0 (+2.0) |

Table 6: Comparison of inference time (ms) per video.

| Method | Pre-Process | Model Inference | Full Pipeline |
|---|---|---|---|
| VideoMAE [42] | 2496.9 | 26.9 | 2523.8 |
| CVPT (Ours) | 238.3 | 25.0 | 263.3 |

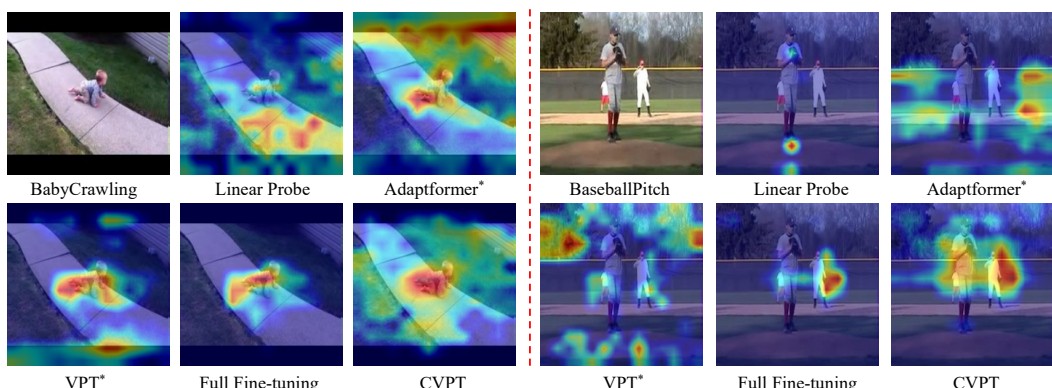

Figure 6: Visualization of the last layer feature map generated by various approaches on UCF-101.

significantly reduces the pre-processing time without the need for video decoding, while maintaining a comparable model inference time, thus remarkably decreasing the overall time cost.

### 4.6 Qualitative Results

We visualize the classification maps generated by CVPT, compared to several state-of-the-art compressed video-based approaches including Linear Probe, Adaptformer*, VPT* and Full Fine-tuning. As shown in Figure 6, our method exhibits a broader focus on motion regions than those compared methods.

## 5 Conclusion and Discussion

We propose a novel prompt tuning method, namely CVPT, for compressed videos. Our approach leverages Motion Vectors and Residuals to generate conditional prompts. Furthermore, we employ the SCCP blocks to fully explore the discriminative dynamic cues in Motion Vectors as complements to I-frames. Extensive experimental results on public benchmarks demonstrate the efficiency and effectiveness of CVPT, compared to the current state-of-the-art approaches for efficient fine-tuning.

Despite the promising performance, the scalability of the proposed CVPT method to various vision tasks such as compressed video-based segmentation and tracking has not been investigated. We will further study this problem in our future research.

## Acknowledgment

This work is partly supported by the National Key R&D Program of China (2021ZD0110503), the National Natural Science Foundation of China (62022011 and 62202034), the Research Program of State Key Laboratory of Software Development Environment, and the Fundamental Research Funds for the Central Universities.

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
