# OpenReview forum: "Compressed Video Prompt Tuning"
_NeurIPS.cc/2023/Conference — NeurIPS 2023 poster_

### Official Review · Reviewer_p8gQ · 2023-06-29

**Soundness:** 2 fair
**Presentation:** 2 fair
**Contribution:** 3 good
**Rating:** 4
**Confidence:** 5

**Summary:**

This work studies the video classification task in the compressed video. With the motion vector and residual as the prompt, this work proposes selective cross-model complementary prompter idea to enhance the cross-model interactions, achieving promising results while maintaining a small number of trainable parameters.

**Strengths:**

Most parts of this paper are well written which clearly demonstrates its motivation and methodology. Especially, the method of this paper is easy to follow.

The idea of using motion vector and residual in the compressed video as prompts is inspiring. These two modalities are computationally free to access, which is a significant advantage compared to traditional motion cues, such as optical flow.

From the experimental results, the proposed CVPT achieves promising results while keeping the tunable parameters low. Visualization is a plus.


**Weaknesses:**

The evaluation benchmarks (UCF, HMDB, and SSv2) are all small scaled. One concern is the scalability of the proposed method.

The design of some key components is not well justified and the figure 2(right) is confusing. For example: 1. Will the prompt join the multi-head self-attention (MHSA)? If so, why the frozen MHSA is able to leverage prompt feature from other modalities. 2. In the SCCP, are the input prompts directly from the previous SCCP other than MHSA? What’s the motivation and effectiveness evidence of keep using the SCCP-processed prompts in SCCP of different layers? 3. In L_2, there is a closed loop for the I-frames embedding. What is the connection to the previous layer. 4. Not sure what are summed together by using the adding function in L_1 and L_2.

The experiments set-up in the ablation study is not clear enough. In the Table 3, CPR without refinement and CPR are both activated in the last row, which is confusing. Also, the linear probe is used as the baseline, but some implementation details are missing, for example: the number of input frames, how prompts attend the self-attention, and the architecture of classifier. In Table 5, what is the implementation details of fully fine-tune?

Ln 45 suggests the large parameter storage burden from previous methods. However, there is no related comparison in the following analysis. In addition to the trainable parameter, the throughput (samples/second), computational burden (GFLOPs), and memory cost (GB) are important metrics.

Minor:

Ln 131 The resulting -> the resulting

There is no notation for g_4 to g_6 in Eq 7

Table 4, E_R -> P_R


**Questions:**

From the Ln124-125, what are the challenges, is there any examples? And is there any evidence to show the inconsistencies between upstream and downstream data?

**Limitations:**

The scalability may be one of the limitations.

---

> ### Author Rebuttal · Authors · 2023-08-09
>
> W1:The evaluation benchmarks (UCF, HMDB, and SSv2) are all small scaled. One concern is the scalability of the proposed method.
>
> A1:In fact, SSv2 is one of the largest datasets used in compressed video, containing a substantial **193,690** videos. Our approach also shows performance improvement on SSv2, illustrating its good scalability.
>
> W2:The design of some key components is not well justified and the figure 2(right) is confusing. For example: 1. Will the prompt join the multi-head self-attention (MHSA)? If so, why the frozen MHSA is able to leverage prompt feature from other modalities. 2. In the SCCP, are the input prompts directly from the previous SCCP other than MHSA? What’s the motivation and effectiveness evidence of keep using the SCCP-processed prompts in SCCP of different layers? 3. In L_2, there is a closed loop for the I-frames embedding. What is the connection to the previous layer. 4. Not sure what are summed together by using the adding function in L_1 and L_2.
>
> A2.1:Prompt will join the MHSA. Despite the frozen MHSA parameters, the capacity to tune both the input visual token and prompts remains intact. Furthermore, our SCCP module effectively functions as modality alignment. This is achieved through the simultaneous integration of improved motion cues into the I-frame token and the concurrent update of the prompt.
>
> A2.2: The SCCP module can dynamically update prompts within each layer to refine the model-tuning process. This strategy is also embraced by the VPT, where new prompts are inserted into each layer. We conducted a comparative experiment, revealing that our strategy yields superior performance in contrast to the alternative method that relies on prompts from the preceding MHSA layer.
> |Method|Acc.|
> |:-|:-:|
> |prompts from MHSA|83.7|
> |Ours|87.6|
>
> A2.3:The I-frame token is sourced from the output of the preceding layer's model output.
>
> A2.4:The adding function employed signifies the summation of $E_I$ from the SCCP output and  I-frame tokens from the backbone.
>
>
> W3:The experiments set-up in the ablation study is not clear enough. In Table 3, CPR without refinement and CPR are both activated in the last row, which is confusing. Also, the linear probe is used as the baseline, but some implementation details are missing, for example: the number of input frames, how prompts attend the self-attention, and the architecture of classifier. In Table 5, what is the implementation details of fully fine-tune?
>
> A3.1:Indeed, we will revise Table 3 to make it more clear.
>
> A3.2:In the linear probe configuration, an equivalent input to the prompt setup is used, encompassing 4 I-frames and 12 P-frames. A uniform embedding function is applied to encode all three modalities. The backbone is pre-trained ViT on the Kinetics-400 with raw videos, and the classification layer is a simple MLP layer that maps the latent features into classification categories. Subsequently, the concatenated tokens across modalities are input into the network. Similar to prior studies [15, 19], only the classification layer is involved in training within the linear probe framework.
>
> A3.3:In the full fine-tuning setup presented in Table 5, the same inputs utilized as in our approach are maintained, consisting of 4 I-frames and 12 P-frames. Distinctive embedding functions are applied to encode the various modalities. The backbone is either ViT or Swin under different pre-training manners. All network parameters are trainable within this configuration.
>
> W4:Ln 45 suggests the large parameter storage burden from previous methods. However, there is no related comparison in the following analysis. In addition to the trainable parameter, the throughput (samples/second), computational burden (GFLOPs), and memory cost (GB) are important metrics.
>
> A4:We make a comprehensive comparison with the raw video method, VideoMAE, in the table below. This comparison effectively demonstrates the superior efficiency of our approach in various aspects. Notably, our method excels in rapid inference by bypassing the need for video decoding. Furthermore, our utilization of compact compressed video inputs contributes to reduced GFLOPs, making our method computationally lighter. Moreover, our innovative prompt tuning framework which only involves training a small subset of the parameter, minimizes memory usage. This presents a significant advantage over the full fine-tuning method.
> |Method|Videos/Second|GFLOPs|Memory Cost (GB)|
> |:-|:-:|:-:|:-:|
> |VideoMAE (ViT-B)|0.40|1080.0|23.6|
> |CVPT (ViT-B)|3.80|772.2|13.4|
>
>
> W5:Ln 131 The resulting -> the resulting, There is no notation for g_4 to g_6 in Eq 7, Table 4, E_R -> P_R
>
> A5:Thank you for your valuable suggestion. We will revise the typo and rectify any potential errors.
>
> Q1:From the Ln124-125, what are the challenges, is there any examples? And is there any evidence to show the inconsistencies between upstream and downstream data?
>
> A1:The challenge under consideration encompasses not only divergences in tasks but also disparities in data modalities. This is evident in scenarios addressed in this paper – transitioning from pre-trained raw video large models to downstream tasks involving compressed video that includes additional motion vectors and residuals. Modality gaps also exist when transitioning from upstream RGB-based tracking models to downstream tasks involving RGB+infrared or RGB+depth-based tracking. Furthermore, this challenge extends to cases where upstream image large models are applied to fine-tuning tasks involving video and text modalities.
>
> L1:The scalability may be one of the limitations.
>
> A1:The progress in compressed video research has led to rapid advancements, despite the relatively smaller dataset scale in comparison to image datasets. Our approach has demonstrated its efficacy on SSv2 dataset which serves as a significant large benchmark. However, due to current constraints, the validation of our method on a larger dataset remains unachievable.

---

> > ### Comment · Reviewer_p8gQ · 2023-08-21
> > **Respond to author rebuttal**
> >
> > Thanks authors for their effort in preparing the rebuttal. After reading through the rebuttal, some of my concerns are solved, such as the confusing tech details and writing part. However, the main concerns are still there for example, the motivation of this work is not well justified, and the scalability .
> > Compared to the K400, the SSv2 is a small scaled dataset in general video understanding benchmarks. Suggest authors to provide more evidence to support the claim in this work.
> > I also try to learn from other reviews who provide accept, however the mentioned strengths were not well supported with evidence. I agree with Reviewer 5c9t, and have similar concerns. As a result, I keep my rating as reject.

---

> > > ### Author Response · Authors · 2023-08-21
> > >
> > > We sincerely appreciate your time and efforts in reviewing our paper.
> > >
> > > Q1: the main concerns are still there for example, the motivation of this work is not well justified, and the scalability . Compared to the K400, the SSv2 is a small scaled dataset in general video understanding benchmarks.
> > >
> > > A1: In compressed video analysis, previous methods have predominantly focused on refining network architectures tailored to align with the characteristics of compressed video. This endeavor inherently involves pre-training these network structures to enhance their performance. Furthermore, some researchers have explored the formulation of self-supervised pre-training tasks meticulously designed for compressed video. Pre-training stands as a pivotal element within these methods, albeit often consuming a significant amount of time. In contrast, our approach introduces an alternative perspective – harnessing prompts to adapt the pre-trained raw video model for tasks involving compressed video. Given the wealth of existing pre-trained raw video models and the inherent strong correlations between raw and compressed videos. Our method sidesteps the need for the pre-training process while achieving results comparable to other compressed video methods.
> > >
> > > SSv2 encompasses a training dataset of 169k instances across 174 categories, placing it within the same order of magnitude as K400. The K400 dataset comprises 240k videos spanning 400 categories, and notably, SSv2 stands out as a challenging dataset that incorporates a greater number of motion-centric action categories. Several studies [40, 6, 44, 39] have presented results on SSv2. We follow their approach, utilizing raw video models pre-trained on K400 in a supervised and self-supervised manner. Our report encompasses results from both smaller datasets (HMDB-51 and UCF-101) and the extensive SSv2 dataset. Furthermore, we intend to include results from K400 using the self-supervised pre-trained models.

---

### Official Review · Reviewer_sECu · 2023-07-03

**Soundness:** 3 good
**Presentation:** 4 excellent
**Contribution:** 3 good
**Rating:** 6
**Confidence:** 3

**Summary:**

The authors present a way to adapt pre-trained raw video models to compressed videos. They utilized the existing concept of prompt tuning from NLP and repurposed it for the compressed video domain. Their findings indicate that by fine-tuning just 0.1 percent of parameters for a downstream task such as video classification, the pre-trained model can be modified to cater to compressed videos.

**Strengths:**

* As per my knowledge, this is the first study to explore the prompt tuning method for the compressed video domain.
* Enough experimental results are provided to back the claims made in paper
* Paper is well written

**Weaknesses:**

* Only video classification is shown as a downstream task.
* Why are the numbers reported in the submission differ from the numbers reported in the original paper(for eg: CoViAR) is there differences in the experimental setting?
* Add Bold text in the tables to highlight the best-performing methods as it is really hard for the readers to sift through the tables in their current condition.

**Questions:**

* Was a LoRA update option considered instead of prompt fine-tuning? If so, can the details of such experiments be mentioned in the supp. material

**Limitations:**

.

---

> ### Author Rebuttal · Authors · 2023-08-09
>
> W1:Only video classification is shown as a downstream task.
>
>
>
> A1:Our designed modules are tailored to address a wide spectrum of tasks pertaining to compressed video. We follow [6] and provide experiments on video classification as a downstream task. Nonetheless, it is necessary in extending our validation to encompass additional tasks. We employed a self-supervised pre-trained ViT-B and subsequently conducted full fine-tuning or prompt tuning on the UCF-101 dataset for video retrieval task. In this context, we extracted the averaged tokens from the last transformer block to serve as the feature representation. the features of the test video queries were matched against the k-nearest-neighbors within the training set features. For assessment, we adopted the recall at k (R@k) metric, consistent with [15, 19]. R@k quantifies the proportion of queries where the top-k nearest neighbors include at least one video belonging to the same class. The experimental results also provide clear evidence of the improvements achieved by our method over full fine-tuning in the context of the retrieval task.
>
> |     Method      | R@1  | R@5  | R@10 | R@20 | R@50 |
> | :-------------: | :--: | :--: | :--: | :--: | :--: |
> | full fine-tune  | 91.4 | 94.3 | 95.3 | 96.5 | 96.7 |
> |    CVPT$^*$     | 87.6 | 90.5 | 92.4 | 95.3 | 96.4 |
> | AdaptFormer$^*$ | 82.3 | 85.3 | 89.9 | 93.7 | 97.0 |
> |      CVPT       | 92.1 | 96.3 | 97.5 | 98.7 | 99.5 |
>
>
>
> W2:Why are the numbers reported in the submission differ from the numbers reported in the original paper(for eg: CoViAR) Are there differences in the experimental setting?
>
>
>
> A2:The results reported for CoViAR in the original paper are grounded under supervised pre-training. In contrast, our study showcases the results under self-supervised pre-training, as reported by IMRNet. Given the prevalence of self-supervised pre-training among the methods being compared, we present CoViAR's results based on self-supervised pre-training, as outlined in the IMRNet. Additionally, we consider incorporating the result of CoViAR under supervised pre-training to enhance the comprehensiveness of our paper.
>
>
>
> W3:Add Bold text in the tables to highlight the best-performing methods as it is really hard for the readers to sift through the tables in their current condition.
>
>
>
> A3:Thanks for your suggestion. We will make improvements in next version.
>
>
>
>
>
> Q1:Was a LoRA update option considered instead of prompt fine-tuning? If so, can the details of such experiments be mentioned in the supp. material
>
>
>
> A1:LoRA introduces an auxiliary module that employs lower-order matrices through mapping weights within the multi-head attention layer. During training, only this auxiliary module is actively trained. This method shares similarities with the practice of enhancing a frozen model with adapters. In contrast to LoRA, the prompt tuning framework offers heightened flexibility in incorporating varying quantities of prompts at diverse positions. Furthermore, this approach simplifies the integration of a priori information into the prompt construction process. While the application of LoRA for compressed video falls outside the scope of this paper, we consider the prospect of exploring this avenue in future endeavors.

---

### Official Review · Reviewer_Eouy · 2023-07-05

**Soundness:** 4 excellent
**Presentation:** 3 good
**Contribution:** 3 good
**Rating:** 8
**Confidence:** 4

**Summary:**

The paper presents one alternative way of finetuning to work on compressed videos. Specifically, it designs a specific data flow within three modalities (RGB, residual, and motion vector). It also presents the way to make the model adapt to new compressed videos and provide a fair comparison. It demonstrates SOTA performance to understand the proposed setting.

**Strengths:**

The problem this paper studies is pretty interesting and useful. Their method design is also complete and thinks about the alternatives.
The presentation is pretty clear while the results are SOTA under their setting.


**Weaknesses:**

The motivation of residual gating motion vector and gating I-Frame information is still weak to me.
It would be better if the author can provide more evidence (exps and visualization).

The presentation of CPR is a little weak to me and needs to be a little more clear.



**Questions:**

Can you also compare the training time & training epochs when you compare with finetune, linear probing, etc?

Can you also compare the data scale between the data used to train the frozen model and the data used for your method training?

If we change the order of motion vector and residual, will the results be better than Fig5b?
(motion vector gating residual and gating I-frame.)

---

> ### Author Rebuttal · Authors · 2023-08-09
>
> W1: The motivation of residual gating motion vector and gating I-Frame information is still weak to me. It would be better if the author can provide more evidence (exps and visualization).
>
> A1:In compressed video, motion vectors capture motion displacement between preceding and subsequent frames via block matching, while residuals represent the remaining error after motion estimation.  Earlier studies [38, 43] have utilized residuals to help rectify erroneous motion information within motion vectors，owing to the coarse and noisy nature of the latter. In this context, we propose the utilization of residuals to generate attention patterns, serving to refine the motion information. We extend the visualization through Figure A, which provides additional insight into the utilization of motion vectors to generate attention for residuals and utilization of residuals to generate attention for I-frames. The experimental results of these strategies are compared below. The generation of attention maps using motion vectors does not exhibit a notable filtering effect when observed through visualization. Consequently, the performance is notably akin to that w/o residual gating.
>
> |            Method             | Acc. |
> | :---------------------------: | :--: |
> |      w/o residual gating      | 86.9 |
> | Motion vector gating residual | 87.0 |
> |             Ours              | 87.6 |
>
>
>
> W2:The presentation of CPR is a little weak to me and needs to be a little more clear.
>
>
>
> A2:Thanks for your suggestion. Every modality within the compressed video exhibits incompleteness and significant inter-modality variations. Simultaneously, the presence of I-frames and the other two modalities is mutually exclusive. Thus, we consider employ distinct embedding functions to encode motion vectors and residuals into separate conditional prompts. Given the relatively longer temporal associated with motion vectors and residuals, an extended temporal tube is utilized to ensure uniform token number across all three modalities. Due to its primary emphasis on edge information within the motion [37], the residual plays a crucial role in enhancing recognition tasks within this specific region. Thus, our approach employs residuals to generate an attention map for filtering motion vectors.
>
>
>
> Q1:Can you also compare the training time & training epochs when you compare with finetune, linear probing, etc?
>
>
>
> A1:we present a comparative analysis of the training time for three distinct methods when conducting 100 epochs on the UCF-101 dataset. Our approach achieves a reduction of over 50% in training time compared to full fine-tuning. Additionally, the training time aligns comparably with the linear probe configuration.
>
>
>
> |     Method     | Training Time | Training Epoch |
> | :------------: | :-----------: | :------------: |
> | full fine-tune |      10h      |      100       |
> |  linear probe  |     4.2h      |      100       |
> |      CVPT      |     4.8h      |      100       |
>
>
>
> Q2:Can you also compare the data scale between the data used to train the frozen model and the data used for your method training?
>
>
>
> A2:The frozen model is pre-trained using the large dataset (Kinetics-400), employing raw videos in both supervised and self-supervised training manners. In contrast, we employed UCF-101, HMDB-51 and SSv2 datasets as distinct training data, respectively.
>
> | Data used for Frozen Model | Data used for our Method |
> | :------------------------: | :----------------------: |
> | Kinetics-400 (240k videos) |  UCF-101 (9.6k videos)   |
> | Kinetics-400 (240k videos) |  HMDB-51 (6.8k videos)   |
> | Kinetics-400 (240k videos) |    SSv2 (169k videos)    |
>
>
>
> Q3:If we change the order of motion vector and residual, will the results be better than Fig5b? (motion vector gating residual and gating I-frame.)
>
>
>
> A3:We present the experimental results of gating residuals with motion vectors in W1 &  A1, yielding a marginal increase of 0.1% compared to the result depicted in Figure 5(b). This finding suggests that the attention generated with motion vectors may not effectively perform the desired filtering, both in terms of visualization and experimental results.

---

### Official Review · Reviewer_5c9t · 2023-07-06

**Soundness:** 4 excellent
**Presentation:** 3 good
**Contribution:** 2 fair
**Rating:** 4
**Confidence:** 4

**Summary:**

This paper proposes an efficient fine-tuning method based on the prompting concept in the compressed video domain. The intuition is to freeze the backbone pre-trained on raw videos and use the proposed prompting techniques to query the required information for the compressed videos. To address the multi-modality of the compressed videos (RGB images, motion vectors, and residuals), the authors propose embedding them first and using a SCCP module to fuse and refine them. The SCCP module is designed based on the fact that the video tasks are more important and sensitive to the motion boundaries. Therefore, the residual map acts as a condition to attend the motion vector map. The results are then added back to the RGB image tokens. The outputs of SCCP are then passed to the next pre-trained layer. Superior performance is obtained and the gap between raw video is shortened.

**Strengths:**

- The idea is simple and easy to follow.
- The efficient design is important to the video tasks as the backbone can be frozen. The overall learnable parameter is significantly small compared to the backbone. Therefore, for different downstream tasks, the storing problem can be alleviated.
- The idea of using the residual map to attend to the motion vectors sounds novel and interesting.

**Weaknesses:**

* Some operations in the proposed method are confusing:
	- In Eq. 6, what is the physical meaning of adding an attended motion vector to image embeddings/tokens? The motion vectors represent the relative movement information for each of the spatial blocks (movement in x and y directions in the form of vectors), while the RGB embedding contains spatial information/structure. The addition operation does not really make sense.
	- The processed \mathcal{M}_l(E)_I^l) is added to E_I^l again, what is the intuition behind this operation? The overall sequence is: the motion vector is attended by the residual information and then added back to the raw image. The output is further processed and added back to the raw image. How to physically explain this?
* The overall architecture is very similar to [51]. What is the fundamental difference?


**Questions:**

L3 should be: existing methods for compressed video classification/application … ?

**Limitations:**

Limitations are addressed.

---

> ### Author Rebuttal · Authors · 2023-08-09
>
> W1:Some operations in the proposed method are confusing:
>
> In Eq. 6, what is the physical meaning of adding an attended motion vector to image embeddings/tokens? The motion vectors represent the relative movement information for each of the spatial blocks (movement in x and y directions in the form of vectors), while the RGB embedding contains spatial information/structure. The addition operation does not really make sense.
>
>
>
> A1:The I-frame tokens are embedded via sparsely sampled I-frame sequence which encompasses both spatial and motion information. However, the intrinsic motion details within these tokens are relatively subdued. To address this limitation, we enhance the I-frame features by incorporating motion vector attributes. This augmentation effectively enriches the motion-related aspects within the I-frame feature representation. Notably, this strategy is commonly employed in both the compressed video domain [42, 45, 28], where RGB and motion information are fused. Although we also considered concatenation as an alternative to addition, we found that it yielded similar performance while incurring higher computational complexity. In raw video domain, where RGB and optical flow information are merged similarly in [52, 53, 54].
>
> [52]. Simonyan K, Zisserman A. Two-stream convolutional networks for action recognition in videos. In NeurIPS, 2014.
>
> [53]. Wang Y, Long M, Wang J, et al. Spatiotemporal pyramid network for video action recognition. In CVPR, 2017.
>
> [54]. Xie D, Deng C, Wang H, et al. Semantic adversarial network with multi-scale pyramid attention for video classification.In AAAI,  2019.
>
>
>
> W2:The processed $\mathcal{M}_l(E)_I^l)$ is added to $E_I^l$ again, what is the intuition behind this operation? The overall sequence is: the motion vector is attended by the residual information and then added back to the raw image. The output is further processed and added back to the raw image. How to physically explain this?
>
>
>
> A2:This operation aimed at incorporating prompts into input tokens, thus facilitating the efficient tuning of the backbone for adaptation to downstream tasks. Within the SCCP module, its objective is to enhance the I-frames feature with essential motion cues. After the SCCP processing, the updated conditional prompts are propagated to the subsequent SCCP layer. Simultaneously, augmented features are assimilated into the input token through an additive mechanism which is a routine operation of the prompt framework.
>
>
>
> W3:The overall architecture is very similar to [51]. What is the fundamental difference?
>
>
>
> A3:Indeed, our method draws inspiration from [51]. Nevertheless, when dealing with the prompt tuning challenge posed by compressed video, a unique complexity arises.  **Firstly**,  each individual modality within compressed video is incomplete and exhibits significant differences among the modalities. Thus, we adopt distinct prompts for encoding motion vectors and residuals within the compressed video. This departure from the unified prompt approach in [1] enhances the efficiency of the entire tuning process. We adopt a similar architecture, as illustrated in Figure 5(a) for comparison. In Table 4, we present a comparative analysis, substantiating how our distinct prompt strategy enhances the efficacy of prompt tuning.  **Secondly**, motion cues in P-frames are coarse and noisy. Leveraging the inter-modality correlation, we introduce a novel operation that purifies motion cues within the SCCP module. This operation facilitates the integration of more accurate motion information into the inputs, consequently bolstering the performance. In contrast, [1] benefits from the utilization of comprehensive and accurate diverse modalities, thereby eliminating the necessity for such refinement.
>
>
>
> Q1:L3 should be: existing methods for compressed video classification/application … ?
>
>
>
> A1:Thank you for your suggestion. We will make modifications to enhance precision and clarity within the text.

---

> > ### Comment · Reviewer_5c9t · 2023-08-19
> > **Respond to author rebuttal**
> >
> > I would like to thank the authors for the rebuttal. However, after reading it, my concerns are not resolved, I decided to retain my rating and vote to rejection, please see the following:
> > * Adding an attended motion vector to I-frame image imbedding is still unclear and inappropriate. The 3 mentioned references are not using similar merging operations:
> >     - [52]: warping is used which is more appropriate and makes sense.
> >     - [53]: mentions that ‘element-wise sum and concatenation do not capture the interactions across the spatial and temporal features, so they may suffer from substantial information loss.’ And they use bilinear fusion operation to model the correlation between each element in spatial and temporal features
> >     - [54]: no interaction operation is mentioned.
> >
> > * The physical meaning is still missing. Why additive mechanism is a routine operation of the prompt framework?
> > * From the authors response, the main difference between [51] is the separately processing of two additional modalities, which may limit the novelty.

---

> > > ### Author Response · Authors · 2023-08-21
> > >
> > > We sincerely appreciate your time and efforts in reviewing our paper.
> > > Q1: Adding an attended motion vector to I-frame image imbedding is still unclear and inappropriate. The 3 mentioned references are not using similar merging operations:
> > >
> > > A1: The comprehensive exploration of spatio-temporal information stands as a fundamental challenge in video representation learning. In regards of compressed videos, I-frames present themselves with sparsely sampled RGB frame sequences from raw videos, encompassing not only spatial characteristics but also partial motion details. However, the limited sampling density results in insufficient representation of motion information, necessitating supplementation through the integration of motion vectors. This process embodies the physical meaning of fusing motion vector embeddings into the I-frame embedding. As for fusion strategies, there are some alternatives in literature besides addition, such as concatenation [45], lateral connection [28] or score-level late fusion [42]. We have already tried these alternatives, and the experimental results indicate their suboptimal efficacy and resource-intensive nature (Utilizing concatenation for fusion in our method leads to an additional 4% increase in parameter count). Therefore, we adopt the simple yet effective addition operation in our submission.
> > >
> > > Regarding [52,53,54], we’d like to make a further clarification. Similarity refers to the fact that within the raw video approach, both the optical flow modality and image embedding, akin to the characteristics of motion vectors, are employed for fusion. Compared to motion vectors, optical flow represents the pixel-wise relative movement information (movement in x and y directions in the form of vectors), resulting in a more intricate and denser flow representation. The specific fusion strategy in [52-54] isn't the primary focus of citing those literatures. Furthermore, we also delve into the fusion strategy adopted by [52-54]. Both [52] and [54] used late fusion at the score level, a fusion that allows the two branches to interact only in the final stage. [27, 28] have indicated that multi-stage feature-level fusion is better than late fusion. [53], on the other hand, uses a better fusion strategy for fusion at the end, but he has a higher computational effort in the fusion stage, which is not suitable for fusion at different stages.
> > >
> > > Q2:The physical meaning is still missing. Why additive mechanism is a routine operation of the prompt framework?
> > >
> > > A2: We will make a more detailed clarification on the physical meaning. $E_I^l$ refers to input data, $\mathcal{M}_l(E)_I^l)$ indicates inserted prompt. By adding $\mathcal{M}_l(E)_I^l)$ into $E_I^l$ the process involves inserting the prompt into the data for fine-tuning. The physical meaning of this process is using the prompt interacts with the input Iframe to adapt the RGB pre-trained model for the downstream compressed video task. In literature, the additive mechanism is widely used in prompt tuning, such as [52], [22], based on which we call it a routine operation.
> > >
> > > Q3: From the authors response, the main difference between [51] is the separately processing of two additional modalities, which may limit the novelty.
> > >
> > > A3: Actually, the main difference is more than separately processing of two additional modalities, and we’d like to make this point clearer. Despite that our work is inspired by [51], our work is substantially different from it in the following aspects. First, [51] is designed for multi-modal tracking for raw videos. In contrast, our work aims to adapt a pre-trained raw video based model to downstream compressed video based vision tasks, which is firstly investigated and extensively evaluated by comparing to existing SOTA approaches in our work. Second, [51] neglects to deal with the core challenge in compressed video, \emph{i.e.} how to integrate the coarse and noisy motion cues in motion vectors and the incomplete spatio-temporal embeddings from I-frames due to sparse sampling. Alternatively, our work develops the SCCP module to specifically address this challenge，by SCCP module which taking in the Iframe token input, as well as the motion vector prompt and the residual prompt. It then generates fused Iframes prompt while simultaneously updating the motion vector and residual prompt. Besides, we also present novel module to refine the coarse and noisy motion vectors by leveraging the residual-based attention. Based on the above difference, as displayed in Table 4, our method achieves significant improvements, with 3.8% on HMDB-51 and 2.1% on UCF-101, compared to [51], clearly showing the effectiveness of the proposed novel components.

---

### Official Review · Reviewer_XoEQ · 2023-07-10

**Soundness:** 3 good
**Presentation:** 3 good
**Contribution:** 2 fair
**Rating:** 6
**Confidence:** 4

**Summary:**

This paper explores how to transfer pretrained RGB models to compressed videos with a parameter-efficient paradigm and introduces a prompt-tuning method named Compressed Video Prompt Tuning (CVPT). In CVPT, the learnable prompts are replaced with encoded compressed modalities that are refined in each layer. To improve cross-modal interactions between prompts and RGB input flow, this paper proposes a Selective Cross-modal Complementary Prompter block that refines the motion cues and complements other modalities to the RGB modality. Experimental results show that CVPT outperforms full fine-finetuning and other prompt-tuning methods on SSv2, UCF101 and HMDB51 dataset.

**Strengths:**

1. The proposed prompt-tuning method designed for compressed videos can leverage the pretrained RGB models with a few
trainable parameters.
2. On compressed videos, CVPT outperforms full fine-finetuning and other prompt-tuning methods (VPT, AdaptFormer) on SSv2, UCF101 and HMDB51 datasets.

**Weaknesses:**

1. The computational cost (GLOPs) of the proposed method may be higher than that of some previous works based on ViT. According to (1), the tokens of I-frames, motion vector prompts and residual prompts are all sent to each layer of pretrained ViT, so the number of input tokens may be larger than that of some previous ViT-based models.
2. This paper says that compressed videos "provide notable advantages in terms of processing efficiency", but there is no efficiency comparison with previous works based on raw videos. It would be better to provide the inference time comparison between previous works (especially RGB models) and the proposed method.
3. Some state-of-the-art methods[2,3] on the UCF101, HMDB51, and SSv2 dataset are not compared in this paper.
4. The idea of this paper is mainly from [1], as mentioned in this paper, but the exploration of compressed video understanding is encouraged.

- [1] Jiawen Zhu, Simiao Lai, Xin Chen, Dong Wang, and Huchuan Lu. Visual prompt multi-modal tracking. CVPR, 2023.
- [2] Christoph Feichtenhofer, Haoqi Fan, Bo Xiong, Ross Girshick, and Kaiming He. A large-scale study on unsupervised spatiotemporal representation learning. CVPR 2021.
- [3] Rui Wang, Dongdong Chen, Zuxuan Wu, Yinpeng Chen, Xiyang Dai, Mengchen Liu, Lu Yuan, and Yu-Gang Jiang. Masked video distillation: Rethinking masked feature modeling for self-supervised video representation learning. CVPR 2023.

**Questions:**

1. The comparison of GLOPs should be provided in Table 1 & Table 2.
2. It would be better to provide the inference time comparison between RGB models and the proposed method.

**Limitations:**

The authors have addressed the limitations of this work.

---

> ### Author Rebuttal · Authors · 2023-08-09
>
> W1&Q1:The computational cost (GLOPs) of the proposed method may be higher than that of some previous works based on ViT. According to (1), the tokens of I-frames, motion vector prompts and residual prompts are all sent to each layer of pre-trained ViT, so the number of input tokens may be larger than that of some previous ViT-based models.  & The comparison of GLOPs should be provided in Table 1 & Table 2.
>
> A1:In our proposed method, while other modalities are employed, each modality is sparsely sampled. In contrast to previous raw video ViT-based approaches (*e.g.* VideoMAE, MotionMAE, MVD), our method yields a reduction in the number of input tokens. Typically, these approaches sample either 16 or 32 frames from a 64 frames sequence with intervals of 4 or 2. In contrast, we sample only 4 I-frames from 4 continuous GOPs, spanning a  48 frames sequence.  In parallel, we selectively extract 3 P-frames (comprising motion vectors and residuals) from each GOP, which serve as the input to our model.  Furthermore, Our method also has reduced tokens compared to previously compressed video ViT-based methods (*e.g.* MM-ViT) which sample the same number of frames to form an input size equivalent to the raw video ViT-based methods.
> Thank you for your suggestion. Due to time constraints, we currently present the GFLOPs of ViT-based approach below. A notable observation emerges that our method has lower GFLOPs compared to other methods due to the reduction in inputs. We intend to provide a comprehensive report of GFLOPs for all methods later.
>
> |Type of Method| Method |GFLOPs|
> | :-: |:-:|:-:|
> |Raw|$M^3$Video|1080.0|
> |Raw|MotionMAE|1080.0|
> |Raw|VideoMAE|1080.0|
> |Raw|MVD-B|1080.0|
> |Raw|VPT|1089.6|
> |Raw|Adaptformer|1093.8|
> |Compressed|CoViAR|1222.0|
> |Compressed|Full Fine-tune|772.2|
> |Compressed|Linear Probe|772.2|
> |Compressed|VPT$^*$|778.2|
> |Compressed|AdaptFormer$^*$|780.6|
> |Compressed|Ours|772.2|
>
> W2&Q2:This paper says that compressed videos "provide notable advantages in terms of processing efficiency", but there is no efficiency comparison with previous works based on raw videos. It would be better to provide the inference time comparison between previous works (especially RGB models) and the proposed method. & It would be better to provide the inference time comparison between RGB models and the proposed method.
>
> A2:Thanks for your suggestion. The efficiency inherent in compressed video analysis manifests through its direct evaluative capacity, obviating the need for video decoding. We present a comprehensive comparison of inference times between our proposed compressed video method and raw video method (*e.g.* VideoMAE) below. Remarkably, our approach demonstrates a significant reduction in pre-processing while maintaining a comparable inference time for model inference.
>
> |Method|Pre-Process (ms)|Model Inference (ms)|Full Pipeline (ms)|
> | :-: | :-: | :-: | :-: |
> |VideoMAE|2496.9|26.9|2523.8|
> |CVPT (Ours)|238.3|25.0|263.3|
>
> W3:Some state-of-the-art methods[2,3] on the UCF101, HMDB51, and SSv2 datasets are not compared in this paper.
>
> A3:We deeply value your suggestion. Both of the two methods are raw video-based methods which leverage a more consistent and comprehensive RGB modality information. Consequently, these approaches exhibit high performance in comparison to our method. Nonetheless, it is noteworthy that their reliance on video decoding, followed by subsequent analysis makes them relatively less efficient in terms of inference time compared to our approach. Furthermore, it is pertinent to highlight that both of these methods necessitate full fine-tuning, whereas our method focuses on tuning only a notably smaller subset of parameters. Finally, we will supplement the comparison with the aforementioned methods by incorporating them into both Table 1 and Table 2.
> |Method|Modality|Model|Input Size [M]|PT Manner|Tunable Params. [M]|HMDB-51|UCF-101|
> |:-:|:-:|:-:|:-:|:-:|:-:|:-:|:-:|
> |MVD-B|RGB|ViT-B|24.1|SSL|86.4 (100%)|76.4| 97.0|
> |$\rho$BYOL|RGB|R3D-50|12.1|SSL|46.9 (100%)|73.6|95.5|
> |Ours|I-Frame+MV+Res|ViT-B|6.1|SSL|0.5 (0.6%)|62.9|89.0|
>
> |Method|Modality|Model|Input Size [M]|PT Manner|Tunable Params. [M]|SSv2|
> |-|-|-|-|-|-|-|
> |MVD-B|RGB|ViT-B|24.1|SSL|86.4 (100%)|72.5|
> |$\rho$BYOL|RGB|R3D-50|12.1|SSL|46.9 (100%)|55.8|
> |Ours|I-Frame+MV+Res|ViT-B|6.1|SSL|0.5 (0.6%)|58.4|
>
> W4:The idea of this paper is mainly from [1], as mentioned in this paper, but the exploration of compressed video understanding is encouraged.
>
> A4:Indeed, our method draws inspiration from [1]. Nevertheless, when dealing with the prompt tuning challenge posed by compressed video, a unique complexity arises. **Firstly**,  each individual modality within compressed video is incomplete and exhibits significant differences among the modalities. Thus, we adopt distinct prompts for encoding motion vectors and residuals within the compressed video. This departure from the unified prompt approach in [1] enhances the efficiency of the entire tuning process. We adopt a similar architecture, as illustrated in Figure 5(a) for comparison. In Table 4, we present a comparative analysis, substantiating how our distinct prompt strategy enhances the efficacy of prompt tuning.  **Secondly**, motion cues in P-frames are coarse and noisy. Leveraging the inter-modality correlation, we introduce a novel operation that purifies motion cues within the SCCP module. This operation facilitates the integration of more accurate motion information into the inputs, consequently bolstering the performance. In contrast, [1] benefits from the utilization of comprehensive and accurate diverse modalities, thereby eliminating the necessity for such refinement.

---

> > ### Comment · Reviewer_XoEQ · 2023-08-21
> >
> > Thanks for authors’ effort in the rebuttal. The responses have addressed most of my questions. However, I noticed that a reviewer has raised doubts about the physical significance of the interaction between motion vector stream and spatial embeddings, and I am also interested in this issue. Overall, I will raise my rate, and I will also adjust my rate based on the answers to the above question. I hope the author incorporates the newly added experimental results and comparisons from the rebuttal into the final version of the paper.

---

> > > ### Author Response · Authors · 2023-08-21
> > >
> > > We sincerely appreciate your time and efforts in reviewing our paper. We intend to enhance the comprehensiveness of our paper by incorporating additional experiments.
> > >
> > > Q1: I noticed that a reviewer has raised doubts about the physical significance of the interaction between motion vector stream and spatial embeddings, and I am also interested in this issue.
> > >
> > > A1: The comprehensive exploration of spatio-temporal information stands as a fundamental challenge in video representation learning. In regards of compressed videos, I-frames present themselves with sparsely sampled RGB frame sequences from raw videos, encompassing not only spatial characteristics but also partial motion details. However, the limited sampling density results in insufficient representation of motion information, necessitating supplementation through the integration of motion vectors. This process embodies the physical meaning of fusing motion vector embeddings into the I-frame embedding.

---

### Author Rebuttal · Authors · 2023-08-10

We are appreciative of the valuable insights shared by the reviewers. In response, we have thoroughly addressed each comment, providing individualized responses to each reviewer. The supplied PDF includes visualizations corresponding to Weakness 1, as referenced by Reviewer Eouy.

---

### Decision · Program_Chairs · 2023-09-21

**Decision:**

Accept (poster)

**Comment:**

This is a borderline submission. The reviewers found the problem relevant and well motivated, and the proposed solution simple and effective. The rebuttal resolved some of the issues raised, with one reviewer left unconvinced. My take is that the simplicity of the method will be interesting for the NeurIPS video understanding community and that the empirical results in this context are strong. Hence, I will recommend acceptance.